# Paclitaxel-Loaded PLGA Coating Stents in the Treatment of Benign Cicatrical Airway Stenosis

**DOI:** 10.3390/jcm11030517

**Published:** 2022-01-20

**Authors:** Xiaojian Qiu, Yan Liu, Jie Zhang, Ting Wang, Juan Wang

**Affiliations:** Department of Respiratory, Beijing Tiantan Hospital, Capital Medical University, No. 119 Southern 4th Ring Road West, Beijing 100070, China; qiuxiaojian@bjtth.org (X.Q.); liuyan12252021@163.com (Y.L.); wangting@bjtth.org (T.W.); wangjuan@bjtth.org (J.W.)

**Keywords:** stent, paclitaxel, benign airway stenosis, treatment

## Abstract

Background: Airway stent implantation used in the treatment of benign cicatricial airway stenosis (BCAS) can lead to local granulation and scar formation, resulting in restenosis and treatment failure. Methods: We systematically investigated a paclitaxel-loaded PLGA-coating stent (PLPCS) and analyzed the safety and efficacy of the PLPCS in patients with BCAS. Patients were enrolled from four hospitals in China and observed for six months after implantation, by bronchoscopy performed weekly in the first month and monthly thereafter. The stent was removed immediately upon detection of granulation tissue proliferation, leading to immobility of the stent. Results: Granulation tissue was formed one week after stent implantation, most of which was located at the upper edge of the stent and the narrowest airway in the stent. All stents were removed in three months (mean: 6.51 + 4.67 weeks), with a curative outcome in one case and ineffective results in two. The remaining seven patients developed complications within three months, necessitating early stent removal. The main complication was granulation formation, resulting in difficulty in stent removal. Conclusion: Although PLPCS showed beneficial effects in basic and animal experiments, it cannot prevent airway restenosis in actual practice, mainly due to granulation formation.

## 1. Introduction

Benign cicatricial airway stenosis (BCAS) is a form of airway stenosis that is caused by proliferation of scar tissue in the trachea, left and right main bronchus, and/or right middle bronchus and results in dyspnea or even suffocation. The treatment has remained challenging. The ideal treatment is surgical removal of the stenosed part. However, if the stenosed portion is too long or inflamed or if the patient is a poor surgical candidate, the treatment may involve bronchoscopy combined with airway stents [1]. However, the continuous compression of the airway by stent and repeated friction between the stent and airway wall during respiration results in injury to the airway mucosa, triggering a local repair reaction and therefore granulation hyperplasia, which in turn can result in airway obstruction [2]. Implantation of silicone stents has been shown to result in various complications such as granulation hyperplasia (49–76%), stent shift (51–70%), and mucus obstruction (17–19%) [3,4]. Metal stents can be easily inserted using a flexible bronchoscope under local anesthesia; in addition, they have good elasticity, can adapt to different types of airways, have good tissue compatibility, and are not likely to shift easily [5,6]. However, granulation proliferation is more serious than that with silicone stents, and their removal after 6–8 weeks of implantation is difficult [7,8,9,10]. Although topical application of antiproliferative drugs such as paclitaxel has been shown to inhibit granulation proliferation in airways, repeated administration of paclitaxel is required to maintain efficiency because it has a short half-life, averaging 5.8 h [11]. Therefore, we hypothesized that coating a metal stent with paclitaxel through Poly(lactic-co-glycolic acid) (PLGA) would allow for the release of the drug from the PLGA coating to the surrounding tissue, thereby increasing its local concentration and duration of action. Additionally, this system would preclude the adverse effects of systemic medication, eliminate the need for repeated administration, and help achieve a curative outcome in BCAS.

Accordingly, we conducted a series of experiments. First, Chen Nan et al. showed that paclitaxel had an inhibitory effect on the proliferation of human pulmonary fibroblasts (HPFs) in vitro. Kong YY successfully reported the development of a coated airway stent with a drug load of 16.3060 ± 0.0021 mg [12]. Wang T et al. showed that at five months after the implantation of the PLPCS in normal dog trachea, the extent of granulation formation was significantly less than that occurring with the bare stent. Combined, these findings suggest that PLPCS can inhibit the formation of scar tissue. In this study, we sought to evaluate the effectiveness and safety of implanting PLPCS in patients with BCAS. As a follow-up to our animal experiments, we aimed to investigate whether the results obtained in our previous studies could be replicated in humans.

## 2. Materials and Methods

### 2.1. Patients

We recruited patients treated for BCAS between 1 June 2014 and 1 June 2016 at any one of the four different hospitals in China, namely, Beijing Tian Tan Hospital (Beijing), Beijing Chest Hospital (Beijing), Qingdao Medical University Affiliated Hospital (Shandong), and Dazhou Central Hospital of Sichuan Province (Sichuan). Including criteria: The included patients were consecutive. The patient met the diagnostic criteria for BCAS [13] and had tracheal lesions, age between 18 and 75 years. Excluding criteria: people who are at increased risk for anesthesia or bronchoscopy. The study was approved by the Ethics Committee of Beijing Tian Tan Hospital, Capital Medical University (JS2013-007-02), and informed consent was obtained from all the patients.

### 2.2. Equipment and Consumables

The following instruments were used for the procedure: (1) rigid bronchoscope (Storz, Tuttlingen, Germany), (2) flexible bronchoscope (BF 240 or 260; Olympus, Tokyo, Japan), (3) jet ventilator (TKR-11300B, Jiangxi Teli Anesthesia Respiratory Equipment Co., Ltd. Nanchang, China), (4) high-frequency electric knife (ERBE, Tuttlingen, Germany), (5) cryotherapy device (ERBE, Tuttlingen, Germany and Kuran, Beijing, China), (6) balloon dilator (Boston Scientific, Boston, MA, USA), and (7) PLPCS (self-developed) [12].

### 2.3. Study Design

This investigation was designed as a single-arm prospective study with only one study group; no control group was included for ethical reasons.

### 2.4. Treatment Methods

The surgery was performed under general anesthesia. First, balloon dilation was performed to dilate the airway. If the scar was too firm for expansion, it was excised using a high-frequency electric knife. Once the stenosis was fully expanded, the base of the scar was subjected to cryotherapy, and the PLPCS was implanted.

### 2.5. Observation Method

The duration of observation was 6 months. Bronchoscopy was performed once per week during the 1st month, and once per month thereafter. At any evaluation, if the scar was found to be embedded in the stent and the stent could not be moved with a forceps, it was immediately removed. At the end of three months, the stent was removed.

### 2.6. Patient Evaluation

The severity of dyspnea was graded using the Modified Medical Research Council Dyspnea scale. Airway stenosis was classified as level 1 if the blockage was at 0–50%, level 2 at 51–70%, level 3 at 71–99%, and level 4 at 100%.

The position of the stenosis was defined as: 1, upper one-third of the trachea; 2, the middle third; and 3, the lower one-third.

Degree of granulation hyperplasia—granuloma at the end of the stent narrowed the lumen diameter: 1, by <25%; 2, by 25~50%, 3, by 50~75%, 4, by more than 75%.

Outcome evaluation: The treatment outcome was considered as (1) curative, if dyspnea index and airway stenosis were improved, and there was no restenosis during the 3-month observation period; (2) improvement, if the dyspnea index and airway stenosis were improved but restenosis occurred within 3 months of observation period; and (3) ineffective, if there was no improvement in the degree of dyspnea or stenosis.

Treatment failure was defined by the removal of the stent before completion of three months due to complications.

Complications: Local complications, such as granulation hyperplasia, mucosal hyperemia, sputum retention, mucosal edema, mucosal necrosis or ulceration, and systemic complications, such as drug-induced allergic reactions, neurotoxicity, hypotension, tachycardia, or gastrointestinal reactions, were observed.

### 2.7. Statistical Analysis

Statistical analysis was performed using the statistical software SPSS 16.5 (IBM, Armonk, NY, USA). Data were represented as mean ± standard deviation. Numerical data were represented by frequency. *p* < 0.05 was considered statistically significant.

## 3. Results

### 3.1. Demographic and Clinical Features of the Patients

Ten patients were enrolled in this study. All patients had BCAS, without any airway malacia (Table 1).

### 3.2. Evaluation of Therapeutic Effect

#### 3.2.1. Immediate Therapeutic Effect

All patients showed immediate improvement after implantation of the stent and significant improvement of the patient’s dyspnea index and airway stenosis (Table 2).

#### 3.2.2. Continuous Therapeutic Effects of PLPCS

##### First Observation after Stent Implant for One Week

Bronchoscopy performed one week after stent implantation showed granulation hyperplasia in all patients (Table 3).

##### Six Months Post-Implantation Follow-Up

Successful removal of the implant at the end of 3 months was possible in 3 patients, where one had a curative outcome, while the remaining two had ineffective outcomes (Table 4). In seven patients, the stents were removed less than 3 months after implantation because of complications, with the average duration of stent implantation being 6.51 ± 4.67 weeks. The main complication was granulation scar hyperplasia. Due to the failure of 70% of the cases, the experiment was terminated in advance, and no cases were reselected.

The findings of the current study indicate that although PLPCSs yield good results in basic research and animal experiments, PLPCS implantation did not show any significant inhibitory effect on the proliferation of the airway scar tissue and airway restenosis [14].

The only patient who showed a curative response was a woman; (Figure 1a). a 14 × 40 mm PLPCS was implanted. (Figure 1b). Granulation hyperplasia occurred within one week of the implantation in the narrowest part of the airway (Figure 1c). We used a biopsy forceps to manipulate the position of the stent; it was shifted up and down by about 0.5 cm every week, for a period of four weeks. Two months after implantation, granulation tissue proliferation in the stent was slightly greater than that before (Figure 1d). At the end of three months, there was no significant increase in the extent of granulation proliferation in the stent (Figure 1e), and it was removed by using a rigid bronchoscope (Figure 1f). The patient was then reexamined monthly for an additional three months (at the end of four, five, and six months), during which there was no further restenosis (Figure 1g–i).

Among the remaining nine cases, three showed treatment effectiveness, while six had treatment failure. The dimensions of the PLPCS implanted in these cases were 16 × 50, 20 × 30, and 16 × 60 mm. However, within one week, granulation hyperplasia had occurred, at the upper edge of the stent in one case, and both at the upper edge and the narrowest part of the airway of the stent in the remaining two cases. Nevertheless, the stent still showed mobility when manipulated with the biopsy forceps (Figure 2a (Case 4)). After 2 weeks, in one case, this stent mobility was not observed, and cryotherapy was required in this case, whereas the two remaining cases required only adjustment with the biopsy forceps. At the 3- and 4-week evaluation, the same methods were used for the management of immobile stents. Five weeks after implantation, only one stent could be moved by the biopsy forceps. Therefore, the immobile stents were removed (Figure 2b (Case 4) and Figure 2 c (Case 8)). In one case, the stent lost its mobility at eight weeks and was then removed. Improvement was noted in the dyspnea index and airway stenosis grade, but the patients still required further treatment, with one patient requiring placement of a silicone stent eventually. Thus, the treatment was effective in these patients (Figure 2d (Case 2)).

The treatment was ineffective in six patients. The sizes of the stents were 18 × 30, 16 × 30, 18 × 40, 16 × 40, 16 × 40, and 16 × 60 mm. Granulation hyperplasia had occurred within one week of implantation in all cases. Mobility on manipulation by the biopsy forceps was seen in four patients at the end of 1 week, three patients at the end of 3 weeks, two patients at the end of 4–8 weeks, and only one patient at the end of 12 weeks. Thus, the average time of stent removal was 7.17 ± 4.67 weeks (2, 3, 4, 10, 12, 12 weeks in the six cases). At the end of six months after operation, no significant improvement was observed in the dyspnea index and airway stenosis grading. Two patients refused further treatment, while the remaining four patients required repeated balloon dilatation; thus, treatment failure was considered in all six cases.

### 3.3. Evaluation of Treatment Safety

#### 3.3.1. Local Complications

The most common local complications of stent implantation are granulation hyperplasia, mucosal hyperemia, and sputum retention; these complications may arise in all patients within one week of implantation. Other local complications include mucosal edema, mucosal necrosis, and ulceration (Table 5).

#### 3.3.2. Systemic Complications

None of the patients had drug-induced allergic reactions, neurotoxicity, hypotension, tachycardia, or gastrointestinal reactions.

#### 3.3.3. Effect of Paclitaxel on Blood Routine, Liver and Kidney Functions

No significant differences were noted between the white blood cell count and serum levels of alanine transaminase, aspartate aminotransferase, and creatinine measured before and one month after treatment (Table 6).

## 4. Discussion

Our findings in this study indicated that PLPCS does not prevent stenosis formation. We were unable to confirm our findings of a beneficial effect of PLPCS in inhibiting granulation formation in humans.

In this study, we did not include a control group with implantation of a bare metal stent due to ethical reasons. Of the ten included patients, a curative outcome was noted in only one case, while improvement was seen in three cases. No improvement was seen in the remaining six patients, which implies that the treatment efficiency in this study was only 40%. This is much less than the 100% rate of effectiveness reported for various other treatment methods for BCAS, such as balloon-dilation, electric knife, cryotherapy, and topical drugs, for membranous and granulation hyperplasia stenosis. Even in complex airway stenosis, the rate of effectiveness reported for other methods is as high as 88% [15]. Therefore, we can infer that PLPCS has no advantage in the treatment of BCAS. In this experiment, only three patients could retain the implant for three months; complications, mainly granulation hyperplasia causing stent immobility, necessitated early removal of the stents in all other cases. The proliferation of granulation tissue may begin from one week of stent implant and increase with the extension of time. The hyperplasia was most marked at the narrowest point of the airway and the upper edge of the stent. We believe that the granulation tissue hyperplasia on the upper edge of the stent may be related to the friction occurring between the stent and the airway wall during respiration and coughing, while that occurring in the middle of the stent may be related to the excessive pressure at the narrowest part. Similar findings were reported by Matsui H et al., who showed that direct contact between the stent wire and the airway wall resulted in inflammatory changes and destruction of the epithelial structure of the airway, resulting in hyperplasia of the scar tissue; they observed that the polyurethane-coated metal stent has low biological activity against the airway epithelial cells and can therefore alleviate foreign body reactions [16]. In another retrospective study, Chung FT et al. reported that patients who underwent stent implantation less than 3 months after lung transplantation were prone to restenosis; they may also have associated local tissue inflammation, which further promotes granulation hyperplasia in the stent [17].

Although in our animal experiments, PLPCS was successfully used to inhibit scar tissue proliferation, the clinical trial for the stent failed to yield the same results [14]. The following may be the reasons for this discrepancy. (1) The airway in the animal experiment was normal, but stenosed in the clinical experiment. This meant that in our patients, the pressure at the middle of stent (stenosis airway) was greater than that in our animal experiments; this increased pressure-caused injury to the airway mucosa, leading to the proliferation of granulation tissue. Therefore, it may be inferred that selecting a stent of the appropriate diameter may help reduce the extent of granulation tissue proliferation in the middle of the stent. (2) We chose the stent manufactured by Nanjing Micro-tech Company. The metallic line of this stent is hard, and the stent diameter will reduce and its length increase when patients cough; this will cause friction between the upper and lower edges of the stent and the airway wall, thereby promoting the proliferation of granulation tissue. However, if we had used the Ultraflex stent, this change in the length of the stent and the subsequent increase in the risk of granulation tissue proliferation will not occur because it is prepared using a different wire-weaving method. (3) In our animal experiments, the airway was not treated with other methods such as an electric knife, cryotherapy, or balloon dilation; however, in the case of clinical trials, such procedures are necessary to expand the stenosed airway before stent implantation. The application of these additional procedures increases the extent of local inflammation, which can themselves trigger granulation tissue proliferation over the long term. Therefore, minimizing airway injury or implanting stents during the period of amelioration of the inflammatory changes in the airway may help reduce granulation proliferation. (4) Paclitaxel not only inhibits scars, but it also inhibits airway mucosal epithelial cells; thus, it affects the epithelialization of stents. Rapamycin does not inhibit airway mucosal epithelial cells and does not affect stent epithelization, but further research is still needed [18]. Our next research direction is the efficacy and safety of rapamycin-coated stents in the treatment of benign cicatricial airway stenosis.

This study has a few limitations. The first is that the study is a single-arm study; no control group was included for ethical reasons. The second is that the number of patients included in this study is ten, which is small. Because the treatment failure was 70%, the experiment was terminated in advance, and no cases were reselected.

In conclusion, the following are the main findings of the current study: (1) PLPCS offers no advantages in the treatment of BCAS. (2) The main reason for treatment failure is granulation hyperplasia occurring at the narrowest part of the airway and upper edge of the stent. (3) The implantation of these stents should be performed during the period of dissipation of airway inflammation in order to minimize airway damage, and stents should be chosen in such a way that the upper and lower edges do not change with the airway diameter; this may help reduce granulation proliferation. (4) Paclitaxel not only inhibits scars, but also inhibits airway mucosal epithelial cells; thus, it affects the epithelialization of stents.

## Figures and Tables

**Figure 1 jcm-11-00517-f001:**
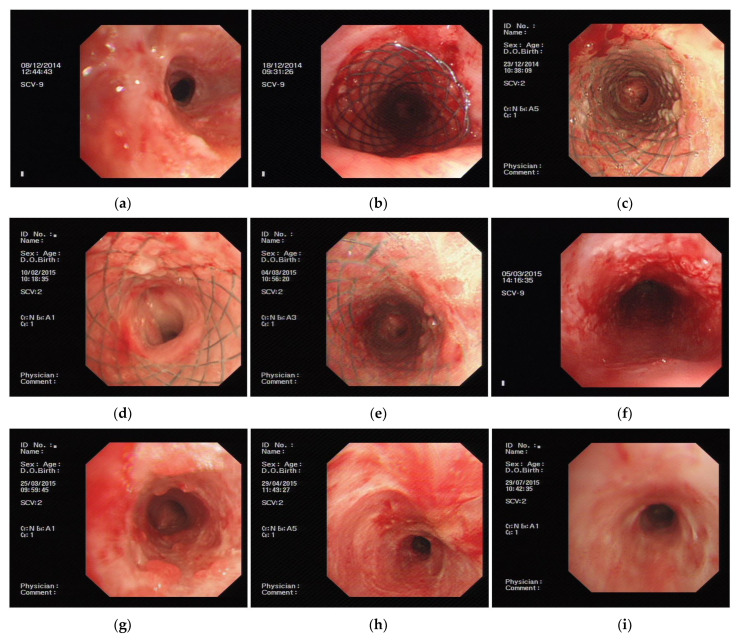
(**a**) Before treatment: tracheal stenosis after tuberculosis. (**b**) After treatment: implant PLPCS. (**c**) One week after stent implant: a small amount of granulation hyperplasia at the narrowest part of the airway. (**d**) Two months after stent implant: granulation hyperplasia was slightly aggravated than before. (**e**) Three months after stent implant: granulation proliferation showed no significant change compared with before. (**f**) After stent was removed: tracheal mucosa was granular and lumen was smooth. (**g**) One month after stent was removed: tracheal mucosa was granular and lumen was smooth. (**h**) Two months after stent was removed: tracheal mucosa was smooth and lumen was smooth. (**i**) Three months after stent was removed: tracheal mucosa was smooth and lumen was smooth.

**Figure 2 jcm-11-00517-f002:**
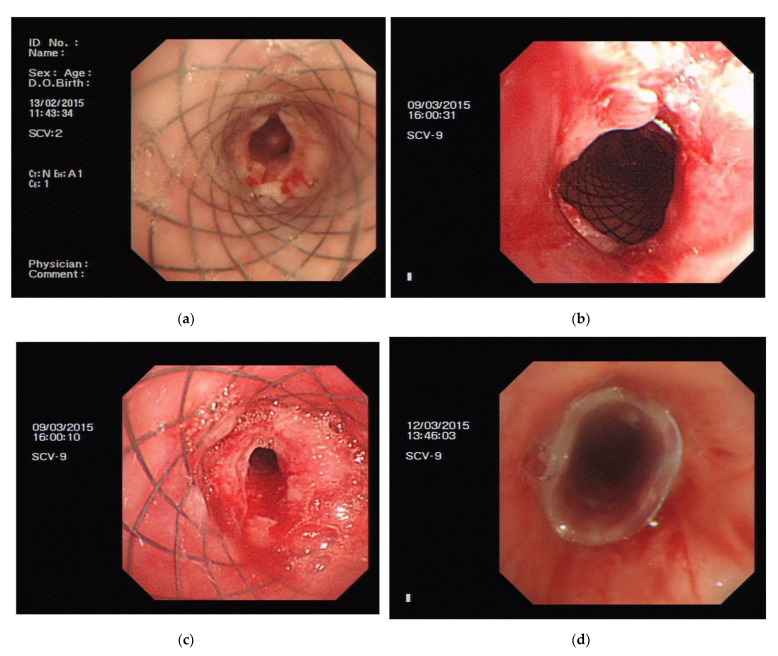
(**a**) Three days after stent implant: granulation hyperplasia at the narrowest part of the airway. (**b**) One month after stent implant: granulation hyperplasia at the upper edge of the stent. (**c**) One month after stent implant: granulation hyperplasia at the narrowest part of the airway. (**d**) Five weeks after stent implant: removed the stent and placed a silicone stent.

**Table 1 jcm-11-00517-t001:** General condition of patients.

Characteristic	Markers	
Gender (case)	male female	5 (50.0%) 5 (50.0%)
Age (years)	41.9 ± 14.43	
Causes	after tuberculosis after tracheal intubation	6 (60.0%) 4 (40.0%)
Dyspnea index	2.70 ± 2.06	
Stenosis degree(case)	1 2 3 4	0 1 9 0
Stenosis length (cm)	2.30 ± 0.79	
Stenosis position (case)	1 2 3	7 2 1

**Table 2 jcm-11-00517-t002:** Changes in patient’s dyspnea index and airway stenosis classification immediately before and after treatment (paired *t*-test).

	Test Group			
	Before	After	*f*	*p*
Dyspnea index	2.70 ± 2.06	0.50 ± 0.71	3.97	0.003
Airway stenosis classification	2.90 ± 0.32	1.10 ± 0.32	13.50	0.000

**Table 3 jcm-11-00517-t003:** Positions of granulation hyperplasia.

Positions	Cases
In the upper edge of stent	2
In the middle of stent	2
All	6

**Table 4 jcm-11-00517-t004:** Continuous observations six months after stent implant.

Case	Gender	Age	Stenosis Conditions				Stent Size	Granulation Hyperplasia Positions	Degree of Granulation Hyperplasia	Epithelialization (Week)	Time of Stent Cannot Move (Week)	Implant Time (Week)	Outcome
			Causes	Positions (Under Glottis) (cm)	Degree	Length (cm)							
1	F	26	1	4.5	3	3.5	14 × 40	2	1	no	/	12	cure
2	M	54	2	1.5	3	3.0	16 × 50	1	1	3	8	8	improvement
3	M	62	2	1.5	2	2.5	18 × 30	1 + 2	1	no	1	2	ineffectiveness
4	M	46	2	1.5	3	2.5	20 × 30	1 + 2	1	no	2	6	improvement
5	F	24	1	0.5	3	1.0	16 × 30	1 + 2	1	no	1	10	ineffectiveness
6	M	62	2	5.5	3	1.5	18 × 40	1	1	no	3	3	ineffectiveness
7	M	41	1	0.5	3	1.5	16 × 40	1 + 2	1	no	4	4	ineffectiveness
8	F	24	1	2.5	3	3.0	16 × 60	1 + 2	1	no	5	5	improvement
9	F	41	1	1.5	3	2.0	16 × 40	2	1	no	12	12	ineffectiveness
10	F	39	1	0.5	3	2.5	16 × 60	1 + 2	1	no	12	12	ineffectiveness

Causes: 1, tuberculosis; 2, tracheal intubation. Stenosis degree: 1, if the blockage was at 0–50%; 2, at 51–70%; 3, at 71–99%; 4, at 100%. Granulation Hyperplasia Positions: 1 upper one-third of the trachea; 2 the middle third; 3 the lower one-third. Degree of granulation hyperplasia: Granuloma narrowed the lumen diameter 1 by <25%; 2 by 25~50%, 3 by 50~75%, 4 more than 75%.

**Table 5 jcm-11-00517-t005:** Local complications.

Case	
Granulation hyperplasia	10
Mucosal hyperemia	10
Sputum retention	10
Mucosal edema	7
Mucosal necrosis or ulceration	6

**Table 6 jcm-11-00517-t006:** Changes in blood routine findings and liver and kidney functions (variance analysis).

	Before Treatment	One Week after Treatment	One Month after Treatment	*f*	*p*
WBC (10^9^/L)	5.15 ± 1.59	5.25 ± 1.46	4.80 ± 0.65	0.201	0.819
ALT (U/L)	15.68 ± 6.82	15.17 ± 5.18	19.83 ± 5.42	1.261	0.303
AST (U/L)	19.27 ± 7.42	17.74 ± 6.61	22.50 ± 3.89	0.982	0.390
Cr (umol/L)	46.48 ± 16.97	50.46 ± 10.12	47.5 ± 6.95	0.236	0.792

*p*: *p* value, *f*: *f* value, WBC: White Blood Cell, ALT: Alanine Transaminase, AST: Aspartate Transaminase.

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
