# Peer review of "Paclitaxel-Loaded PLGA Coating Stents in the Treatment of Benign Cicatrical Airway Stenosis"

_jcm, 2022, doi:10.3390/jcm11030517_

Round 1
Reviewer 1 Report
I want to thank the handling editor for giving me the opportunity to review this interesting study. I would also like to congratulate the authors for their elegant manuscript. Here, I have made some suggestions that, in my opinion, could help improve the overall quality of the manuscript.
Title
- The authors may consider providing the abbreviations in full to allow for better understanding by the reader who is not familiar with the terms.
Introduction
- Line 31: The authors may consider using a more accurate term instead of “unaffordable for surgery” (e.g., poor surgical candidate).
Materials and Methods
- The authors may consider clarifying the nature of the study (retrospective versus prospective).
- The authors may consider stating if the included patients were consecutive.
- Line 80: The authors may consider replacing the term “surgery” with a more accurate one (e.g., rigid bronchoscopy).
- Line 102: The authors may consider naming for which complications they collected data.
Discussion
- The authors may consider reporting and analysing the limitations of their study.
- The 4th (last) point of the conclusions may be valid but does not appear to be a logical consequence of the findings of the present study.
Author Response
|
Reviewer Number |
Original comments of the reviewer |
Reply by the author(s) |
Changes done on page number and line number |
|
1 |
Title The authors may consider providing the abbreviations in full to allow for better understanding by the reader who is not familiar with the terms.
|
Thanks, I have changed the topic. Paclitaxel-loaded PLGA Coating Stents in the Treatment of Benign Cicatricial Airway Stenosis. |
1 |
|
|
Introduction Line 31: The authors may consider using a more accurate term instead of “unaffordable for surgery” (e.g., poor surgical candidate).
|
Thanks, I have changed to unaffordable for surgery. |
54 |
|
|
Materials and Methods The authors may consider clarifying the nature of the study (retrospective versus prospective).
|
Thanks, I have changed. |
101 |
|
|
The authors may consider stating if the included patients were consecutive.
|
Thanks, I have added. |
89 |
|
|
Line 80: The authors may consider replacing the term “surgery” with a more accurate one (e.g., rigid bronchoscopy).
|
Thanks. We don’t always use rigid bronchoscopy, laryngeal mask was also used.
|
99 |
|
|
Line 102: The authors may consider naming for which complications they collected data. |
Thanks, I have added. |
123 |
|
|
Discussion The authors may consider reporting and analysing the limitations of their study.
|
Thanks, I have added. |
287 |
|
|
The 4th (last) point of the conclusions may be valid but does not appear to be a logical consequence of the findings of the present study.
|
Thanks, I have changed. |
297 |

Reviewer 2 Report
This is the challenging report that new treatment method of benign cicatricial airway stenosis (BCAS).
Enough effects do not seem to have been obtained with paclitaxel this time, but expect it to future rapamycin so that the authors consider it.
It is useful and I think that it was able to be a valuable report by major revision. Here is a summary of steps that I would recommend:
Major revision:
1. Please write it on the text(P3 L115) or the table 2 about the statistics method that you used. Is this the paired T-test?
2. Please write it on the text(P7 L186) or the table 6 about the statistics method that you used. Is it the P value that compared only "Before treatment" with "One month after treatment" so that described in the manuscript? We are thankful when you can add such a comment to Tabel 6 if so.
3. It is time series of figure 1, but thinks Fig. 1-7 to be three weeks later from a stent removal exactly.
I think that this is permitted on account of the protocol and the due date of the examination.
However, please revise Fig 1-9 from a stent removal three months later for such as "five months".
4. What is "Degree of granulation hyperplasia" in Table 4 ? And there is no title of Table 4.
5. Figure 1 was found to be the time series of patients of Case 1, but it was the photograph of patients of which Case No, respectively, or figure 2 was incomprehensible from the text. Would you please modify "One patient" to such as "Case 2" of the manuscript, and put notation for example "Case 2" in the each figure 2?
Minor revision:
1. P1 L10 Please revise "cicatrical" for "cicatricial".
2. P1 L27 Please revise "cicatrical" for "cicatricial".
3. P1 L30 Please delete the redundant period.
4. P1 L41 Please revise "seriousthan" for "serious than".
5. P2 L46 At first, please describe the explanation of the abbreviation PLGA such as "Poly lactic acid-co-glycolic acid (PLGA)".
6. P2 L55 Please revise "PCPLS" for "PLPCS".
7. P2 L56 It may be necessary for you to add references [12] to the end.
8. P2 L88 Please add the period.
9. P4 L133 A format of the quotation becomes the cock-up from the twelfth. Please standardize it.
10. P4 L136 Notation of "Fig." and "Figure "is mixed. Please standardize it.
11. P7 Tabel 6 Please revise "22.503.89".
12. P7 Tabel 6 Please add a unit to each item.
13. P8 L253 Is "BAS" means "BCAS"?
14. Please add 1-1, 1-2, ... 2-1, 2-2 at upper left each figure.
15. "*" is mixed with "x" about a mention method of the stent size, and please standardize it in manuscripts and table.
Author Response
|
Reviewer Number |
Original comments of the reviewer |
Reply by the author(s) |
Changes done on page number and line number |
|
2 |
Major revision: Please write it on the text(P3 L115) or the table 2 about the statistics method that you used. Is this the paired T-test?
|
Thanks, I used paired T-test. |
144 |
|
|
2. Please write it on the text(P7 L186) or the table 6 about the statistics method that you used. Is it the P value that compared only "Before treatment" with "One month after treatment" so that described in the manuscript? We are thankful when you can add such a comment to Tabel 6 if so.
|
Thanks, I used variance analysis and I have added on the topic of Table 6. |
230 |
|
|
3. It is time series of figure 1, but thinks Fig. 1-7 to be three weeks later from a stent removal exactly.
|
Thanks, I have added this sentence: at the end of four, five, six month. |
178 |
|
|
4. What is "Degree of granulation hyperplasia" in Table 4 ? And there is no title of Table 4.
|
I have added degree of granulation hyperplasia: Granuloma at the end of the stent narrowed the lumen diameter 1 by < 25%; 2 by 25% ~ 50%, 3 by 50% ~ 75%, 4 more than 75%. Title of Table 4: Continuous observations of six months after stent implant |
114
161 |
|
|
5. Figure 1 was found to be the time series of patients of Case 1, but it was the photograph of patients of which Case No, respectively, or figure 2 was incomprehensible from the text. Would you please modify "One patient" to such as "Case 2" of the manuscript, and put notation for example "Case 2" in the each figure 2?
|
Thanks, I have added. Fig. 2-1(Case 4), Fig. 2-2(Case 4), 2-3(Case 8), Fig. 2-4(Case 2).
|
189 194 198 |
|
|
Minor revision: |
Thanks, I have modified. |
28 |
|
|
2. P1 L27 Please revise "cicatrical" for "cicatricial".
|
Thanks, I have modified. |
50 |
|
|
3. P1 L30 Please delete the redundant period.
|
Thanks, I have modified. |
53 |
|
|
4. P1 L41 Please revise "seriousthan" for "serious than".
|
Thanks, I have modified. |
64 |
|
|
5. P2 L46 At first, please describe the explanation of the abbreviation PLGA such as "Poly lactic acid-co-glycolic acid (PLGA)".
|
Thanks, I have modified. |
69 |
|
|
6. P2 L55 Please revise "PCPLS" for "PLPCS".
|
Thanks, I have modified. |
78 |
|
|
7. P2 L56 It may be necessary for you to add references [12] to the end.
|
Thanks, I have modified. |
168 |
|
|
8. P2 L88 Please add the period.
|
Thanks, I have modified. |
109 |
|
|
9. P4 L133 A format of the quotation becomes the cock-up from the twelfth. Please standardize it.
|
Thanks, I have modified. |
168 |
|
|
10. P4 L136 Notation of "Fig." and "Figure "is mixed. Please standardize it.
|
Thanks, I have modified. |
170-179 |
|
|
11. P7 Tabel 6 Please revise "22.503.89".
|
Thanks, I have modified. |
230 |
|
|
12. P7 Tabel 6 Please add a unit to each item.
|
Thanks, I have modified. |
230 |
|
|
13. P8 L253 Is "BAS" means "BCAS"?
|
Thanks, I have deleted. |
298 |
|
|
14. Please add 1-1, 1-2, ... 2-1, 2-2 at upper left each figure.
|
Thanks, I have modified. |
181,198 |
|
|
15. "*" is mixed with "x" about a mention method of the stent size, and please standardize it in manuscripts and table. |
Thanks, I have modified. |
161 |

Reviewer 3 Report
In general, this manuscript on 10 cases is well written regarding scientific aspects and format.
My comments:
- English requires improvement.
- Abstract: "One week after implantation, all patients had granulation formation, mostly on the upper edge of the stent and the narrowest airway in the stent." This sentence should be corrected.
- Discussion. "Rapamycin coated airway stents would be the ideal choice in the treatment of BAS and further studies are required for their development." This sentence cannot be a conclusion of this case series as Rapamycin-coated stent is not used in any of the cases. It can only be a comment in the Discussion and there should be a reference or references for rapamycin if the authors are mentioning about its effectiveness.
Author Response
|
Reviewer Number |
Original comments of the reviewer |
Reply by the author(s) |
Changes done on page number and line number |
|
3 |
1.English requires improvement.
|
Thanks, I have modified. |
|
|
|
2.Abstract: "One week after implantation, all patients had granulation formation, mostly on the upper edge of the stent and the narrowest airway in the stent." This sentence should be corrected.
|
Thanks, I have modified. |
37 |
|
|
3.Discussion. "Rapamycin coated airway stents would be the ideal choice in the treatment of BAS and further studies are required for their development." This sentence cannot be a conclusion of this case series as Rapamycin-coated stent is not used in any of the cases. It can only be a comment in the Discussion and there should be a reference or references for rapamycin if the authors are mentioning about its effectiveness.
|
Thanks, I have deleted this sentence and have added a reference. |
284 |

Reviewer 4 Report
Dear Authors thanks for this very interesting study which demonstrates that even with placlitaxel loaded PLGA coating metallic stent leading to a lot of complication on benign airway stenosis.
however I have few comments :
Major comments :
1) in the Discussion part more comparaison between silicon and other study using metallic stent seems necessary
2) in an other paragraph authors compare two metallic stent ( Microthec and Boston ) , this comparison need more explanation and references.
3) on the last paragraph authors should develop the paragraph with Rapamicin and add references , moreover this study could not conclude that "Rapamicin coated airway stents would be the ideal choice" : this is not the point of this study.
Minor comments
1) on the tittle : maybe authors should replace BCAS abbreviation, and in backround , authors could more explain what is PLGA.
2) on the introduction please add more references for metallic stent study for benign stenosis
3) in material and methods : name of hospital is missing and replace by xxx
4) in the discussion : please add references line 218 " although in our animal experiment..."
Author Response
|
Reviewer Number |
Original comments of the reviewer |
Reply by the author(s) |
Changes done on page number and line number |
|
4 |
Major comments : 1) in the Discussion part more comparaison between silicon and other study using metallic stent seems necessary
|
Thanks. In the introduction, comparaison between silicon and metallic stent has been mentioned. The details are as follows: Implantation of silicone stents has been shown to result in various complications such as granulation hyperplasia (49%-76%), stent shift (51%-70%), mucus obstruction (17%-19%) [3,4]. Metal stents can be easily inserted using a flexible bronchoscope and under local anesthesia; also, they have good elasticity, can adapt to different types of airways, have good tissue compatibility, and are not likely to shift easily. |
|
|
|
2) in another paragraph authors compare two metallic stent (Microthec and Boston ) , this comparison need more explanation and references.
|
Thanks. At present, there is no articles comparing the two stents (Microthec and Boston ) . The comparison of the two stents is one of the innovations of my article. By comparing the similarities and differences of the two stents, we can find the stents with more convenient operation and less damage to patients. |
|
|
|
3) on the last paragraph authors should develop the paragraph with Rapamicin and add references , moreover this study could not conclude that "Rapamicin coated airway stents would be the ideal choice" : this is not the point of this study.
|
Thanks, I have deleted this sentence and have added a reference. |
284 |
|
|
Minor comments 1) on the tittle : maybe authors should replace BCAS abbreviation, and in backround , authors could more explain what is PLGA.
|
Thanks, I have modified. |
1 69 |
|
|
2) on the introduction please add more references for metallic stent study for benign stenosis
|
Thanks, I have added references. |
63 335-342 |
|
|
3) in material and methods : name of hospital is missing and replace by xxx
|
This was done by the editor, who may be afraid that the experts will see which hospital it is and affect the review results. |
|
|
|
4) in the discussion : please add references line 218 " although in our animal experiment..."
|
Thanks, I have added. |
262 |

Round 2
Reviewer 2 Report
Thank you very much for correcting politely. I think one at the figure became very easy to look in in particular. It's expected of future's further study.